# One-Shot Unsupervised Cross Domain Translation

Sagie Benaim[1] and Lior Wolf[1,2]

[1]The School of Computer Science , Tel Aviv University, Israel
[2]Facebook AI Research

## Abstract

Given a single image $x$ from domain $A$ and a set of images from domain $B$, our task is to generate the analogous of $x$ in $B$. We argue that this task could be a key AI capability that underlines the ability of cognitive agents to act in the world and present empirical evidence that the existing unsupervised domain translation methods fail on this task. Our method follows a two step process. First, a variational autoencoder for domain $B$ is trained. Then, given the new sample $x$, we create a variational autoencoder for domain $A$ by adapting the layers that are close to the image in order to directly fit $x$, and only indirectly adapt the other layers. Our experiments indicate that the new method does as well, when trained on one sample $x$, as the existing domain transfer methods, when these enjoy a multitude of training samples from domain $A$. Our code is made publicly available at `https://github.com/sagiebenaim/OneShotTranslation`.

## 1   Introduction

A simplification of an intuitive paradigm for accumulating knowledge by an intelligent agent is as follows. The gained knowledge is captured by a model that retains previously seen samples and is also able to generate new samples by blending the observed ones. The agent learns continuously by being exposed to a series of objects. Whenever a new sample is observed, the agent generates, using the internal model, a virtual sample that is analogous to the observed one, and compares the observed and blended objects in order to update the internal model.

This variant of the perceptual blending framework [1], requires multiple algorithmic solutions. One major challenge is a specific case of "the learning paradox", i.e., how can one learn what it does not already know, or, in the paradigm above, how can the analogous mental image be constructed if the observed sample is unseen and potentially very different than anything that was already observed.

Computationally, this generation step requires solving the task that we term one-shot unsupervised cross domain translation: given a single sample $x$ from an unknown domain $A$ and many samples or, almost equivalently, a model of domain $B$, generate a sample $y \in B$ that is analogous to $x$. While there has been a great deal of research dedicated to unsupervised domain translation, where many samples from domain $A$ are provided, the literature does not deal, as far as we know, with the one-shot case.

To be clear, since parts of the literature may refer to these type of tasks as zero-shot learning, we are not given any training images in $A$ except for the image to be mapped $x$. Consider, for example, the task posed in [2] of mapping zebras to horses. The existing methods can perform this task well, given many training images of zebras and horses. However, it seems entirely possible to map a single zebra image to the analogous horse image even without seeing any other zebra image.

The method we present, called OST (One Shot Translation), uses the two domains asymmetrically and employs two steps. First, a variational autoencoder is constructed for domain $B$. This allows us to encode samples from domain $B$ effectively as well as generate new samples based on random

latent space vectors. In order to encourage generality, we further augment $B$ with samples produced by a slight rotation and with a random horizontal translation.

In the second phase, the variational autoencoder is cloned to create two copies that share the top layers of the encoders and the bottom layers of the decoders, one for the samples in $B$ and one for the sample $x$ in $A$. The autoencoders are trained with reconstruction losses as well as with a single-sample one-way circularity loss. The samples from domain $B$ continue to train its own copy as in the first step, updating both the shared and the unshared layers. The gradient from sample $x$ updates only the unshared layers and not the shared layers. This way, the autoencoder of $B$ is adjusted by $x$ through the loss incurred on unshared layers for domain $B$ by the circularity loss, and through subsequent adaptation of the shared layers to fit the samples of $B$. This allows the shared layers to gradually adapt to the new sample $x$, but prevents overfitting on this single sample. Augmentation is applied, as before, to $B$ and also to $x$ for added stability.

We perform a wide variety of experiments and demonstrate that OST outperforms the existing algorithms in the low-shot scenario. On most datasets the method also presents a comparable accuracy with a single training example to the accuracy obtained by the other methods for the entire set of domain $A$ images. This success sheds new light on the potential mechanisms that underlie unsupervised domain translation, since in the one-shot case, constraints on the inter-sample correlations in domain $A$ do not apply.

## 2   Previous Work

Unsupervised domain translation methods receive two sets of samples, one from each domain, and learn a function that maps between a sample in one domain and the analogous sample in the other domain [2, 3, 4, 5, 6, 7, 8, 9, 10, 11, 12]. Such methods are unsupervised in the sense that the two sets are completely unpaired.

The mapping between the domains can be recovered based on multiple cues. First, shared objects between domains can serve as supervised samples. This is the case in the early unsupervised cross-lingual dictionary translation methods [13, 14, 15, 16], which identified international words ('computer', 'computadora','kompüter') or other words with a shared etymology by considering inter-language edit distances. These words were used as a seed set to bootstrap the mapping process.

A second cue is that of object relations. It often holds that the pairwise similarities between objects in domain $A$ are preserved after the transformation to domain $B$. This was exploited in [5] using the L2 distances between classes. In the work on unsupervised word to word translation [9, 10, 11, 17], the relations between words in each language are encoded by word vectors [18], and translation is well approximated by a linear transformation of one language's vectors to those of the second.

A third cue is that of inner object relations. If the objects of domain $A$ are complex and contain multiple parts, then one can expect that after mapping, the counterpart in domain $B$ would have a similar arrangement of parts. This was demonstrated by examining the distance between halves of images in [5] and it also underlies unsupervised NLP translation methods that can translate a sentence in one language to a sentence in another, after observing unmatched corpora [12].

Another way to capture these inner-object relations is by constructing separate autoencoders for the two domains, which share many of the weights [6, 7]. It is assumed that the low-level image properties, such as texture and color, are domain-specific, and that the mid- and top-level properties are common to both domains.

The third cue is also manifested implicitly (in both autoencoder architectures and in other methods) by the structure of the neural network used to perform the cross-domain mapping [19]. The network's capacity constrains the space of possible solutions and the relatively shallow networks used, and their architecture dictate the form of a solution. Taken together with the GAN [20] constraints that ensure that the generated images are from the target domain, and restricted further by the circularity constraint [2, 3, 4], much of the ambiguity in mapping is eliminated.

In the context of one-shot translation, it is not possible to find or to generate analogs in $B$ to the given $x \in A$, since the domain-invariant distance between the two domains is not defined. One can try to use general purpose distances such as the perceptual distance, but this would make the work

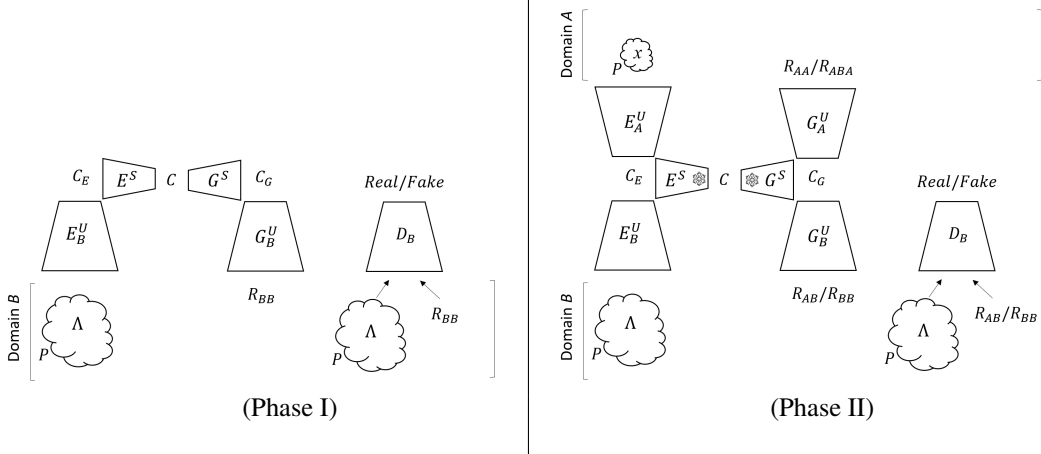

Figure 1: Illustration of the two phases of training. (Phase I): Augmented samples from domain $B$, $P(\Lambda)$, are used to train a variational autoencoder for domain $B$. $R_{BB}$ denotes the space of reconstructed samples from $P(\Lambda)$. (Phase II): the variational autoencoder of phase I is cloned, while sharing the weights of part of the encoder ($E^S$) and part of the decoder ($G^S$). These shared parts, marked with a snowflake, are frozen with respect to the sample $x$. For both phase I and phase II, we train a discriminator $D_B$ to ensure that the generated image belong to the distribution of domain $B$. $P(x)$ and $P(\Lambda)$ are translated to a common feature space, $C_E$, using $E_A^U$ and $E_B^U$ respectively. $C$ (resp $C_G$) is the space of features, constructed after passing $C_E$ (resp $C$) through the common encoder $E^S$ (resp common decoder $G^S$). $R_{AB}$ denotes the subspace of samples in $B$ constructed from $P(x)$, which is generated by augmenting $x$. $R_{AA}$ denotes the space of reconstructed samples from $P(x)$. $R_{ABA}$ denotes the subspace of samples in $A$ constructed by translating $P(x)$ to domain $B$ and then back to $A$.

semi-supervised such as [21, 22] (these methods are also not one-shot). Since there are no inter-object relations in domain $A$, the only cue that can be used is of the third type.

We have made attempts to compare various image parts within $x$, thereby generalizing the image-halves solution of [5]. However, this did not work. Instead, our work relies on the assumption that the mid-level representation of domain $A$ is similar to that of $B$, which, as mentioned above, is the underlying assumption in autoencoder based cross-domain translation work [6, 7].

## 3 One-Shot Translation

In the problem of unsupervised cross-domain translation, the learning algorithm is provided with unlabeled datasets from two domains, $A$ and $B$. The goal is to learn a function $T$, which maps samples in domain $A$ to analog samples in domain $B$. In the autoencoder based mapping technique [7], two encoders/decoders are learned. We denote the encoder for domain $A$ ($B$) by $E_A$ ($E_B$) and the decoder by $G_A$ ($G_B$). In order to translate a sample $x$ in domain $A$ to domain $B$, one employs the encoder of $A$ and the decoder of $B$, i.e., $T_{AB} = G_B \circ E_A$.

A strong constraint on the form of the translation is given by sharing layers between the two autoencoders. The lower layers of the encoder and the top layers of the decoder are domain-specific and unshared. The encoder's top layers and decoder's bottom layers are shared. This sharing enforces the same structure on the encoding of both domains and is crucial for the success of the translation.

Specifically, we write $E_A = E^S \circ E_A^U$, $E_B = E^S \circ E_B^U$, $G_A = G_A^U \circ G^S$, and $G_B = G_B^U \circ G^S$, where the superscripts $S$ and $U$ denote shared and unshared parts, respectively, and the subscripts denote the domain. This structure is depicted in Fig. 1.

In addition to the networks that participate in the two autoencoders, an adversarial discriminator $D_B$ is trained in both phases, in order to model domain $B$. Domain $A$ does not contain enough real examples in the case of low-shot learning and, in addition, a domain $A$ discriminator is less needed since the task is to map from $A$ to $B$. When mapping $x$ (after augmentation, to $B$ using the transformation $T$) the discriminator $D_B$ is used to provide an adversarial signal.

### 3.1 Phase One of Training

In the first phase, we employ a training set $\Lambda$ of images from domain B and train a variational autoencoder for this domain. The method employs an augmentation operator that consists of small random rotations of the image and a horizontal translation. We denote by $P(\Lambda)$ the training set constructed by randomly augmenting every sample $s \in \Lambda$.

The following losses are used:

$$\mathcal{L}_{REC_B} = \sum_{s \in P(\Lambda)} \|G_B(E_B(s)) - s\|_1 \tag{1}$$

$$\mathcal{L}_{VAE_B} = \sum_{s \in P(\Lambda)} \text{KL}(E_B \circ P(\Lambda) \| \mathcal{N}(0, I)) \tag{2}$$

$$\mathcal{L}_{\text{GAN}_B} = \sum_{s \in P(\Lambda)} -\ell(\overline{D_B}(G_B(E_B(s))), 0) \tag{3}$$

$$\mathcal{L}_{\text{D}_B} = \sum_{s \in P(\Lambda)} +\ell(D_B(\overline{G_B}(\overline{E_B}(s))), 0) + \ell(D_B(s), 1) \tag{4}$$

where the first three losses are the reconstruction loss, the variational loss and the adversarial loss on the generator, respectively, and the fourth loss is the loss of the GAN's discriminator, in which we use the bar to indicate that $G_B$ is not updated during the backpropagation of this loss. $\ell$ can be the binary cross entropy or the least square loss, $\ell(x, y) = (x - y)^2$ [23]. When training $E_B$ and $G_B$ in the first phase, the following loss is minimized:

$$\mathcal{L}^I = \mathcal{L}_{REC_B} + \alpha_1 \mathcal{L}_{VAE_B} + \alpha_2 L_{\text{GAN}} \tag{5}$$

where $\alpha_i$ are tradeoff parameters. At the same time we train $D_B$ to minimize $\mathcal{L}_{D_B}$. Similarly to CycleGAN, $D_B$ can be a PatchGAN [24] discriminator, which checks if $70 \times 70$ overlapping patches of the image are real or fake.

### 3.2 Phase Two of Training

In the second phase, we make use of the sample $x$ from domain $A$, as well as the set $\Lambda$. In case we are given more than one sample from domain $A$, we simply add the loss terms to each one of the samples.

Denote by $P(x)$ the set of random augmentations of $x$ and the cross-domain encoding/decoding as:

$$T_{BB} = G_B^U(\overline{G^S}(\overline{E^S}(E_B^U(x)))) \tag{6} \qquad T_{AA} = G_A^U(\overline{G^S}(\overline{E^S}(E_A^U(x)))) \tag{8}$$

$$T_{BA} = G_A^U(\overline{G^S}(\overline{E^S}(E_B^U(x)))) \tag{7} \qquad T_{AB} = G_B^U(\overline{G^S}(\overline{E^S}(E_A^U(x)))) \tag{9}$$

where the bar is used, as before, to indicate a detached clone not updated during backpropagation. $G_B^U$ and $G_A^U$ (resp. $E_B^U$ and $E_A^U$) are initialized with the weights of $G_A$ (resp. $E_A$) trained in phase I.

The following additional losses are used:

$$\mathcal{L}_{REC_A} = \sum_{s \in P(x)} \|T_{AA}(s) - s\|_1 \tag{10}$$

$$\mathcal{L}_{\text{cycle}} = \sum_{s \in P(x)} \|T_{BA}(T_{AB}(s)) - s\|_1 \tag{11}$$

$$\mathcal{L}_{\text{GAN}_{AB}} = \sum_{s \in P(x)} -\ell(\overline{D_B}(T_{AB}(s)), 0) \tag{12}$$

$$\mathcal{L}_{\text{D}_{AB}} = \sum_{s \in P(x)} +\ell(D_B(\overline{T_{AB}}(s)), 0) + \ell(D_B(s), 1) \tag{13}$$

namely, the reconstruction loss on $x$, a one-way cycle loss applied to $x$, and the generator and discriminator losses for domain $B$ given the source sample $x$. In phase II we minimize the following loss:

$$\mathcal{L}^{II} = \mathcal{L}^I + \alpha_3 \mathcal{L}_{REC_A} + \alpha_4 \mathcal{L}_{\text{cycle}} + \alpha_5 \mathcal{L}_{\text{GAN\_AB}} \tag{14}$$

where $\alpha_i$ are tradeoff parameters. Losses not in $\mathcal{L}^I$ are minimized over the unshared layers of the encoders and decoders. We stress that losses in $\mathcal{L}^I$ as still minimized over both the shared and unshared layers in phase II. At the same time we train $D_B$ to minimize $\mathcal{L}_{D_B}$ and $\mathcal{L}_{D_{AB}}$.

Note that $G^S$ and $E^S$ enforce the same structure on $x$ as it does on samples from domain $B$. Enforcing this is crucial in making $x$ and $T_{AB}(x)$ structurally aligned, as these layers typically encode structure common to both domains $A$ and $B$ [7, 6]. OST assumes that it is sufficient to train a VAE for domain $B$ only, in order for $G^S$ and $E^S$ to contain the features needed to represent $x$ and its aligned counterpart $T_{AB}(x)$. Give this assumption, it does not rely on samples from $A$ to train $G^S$ and $E^S$.

$G^S$ and $E^S$ are detached during backpropagation not just from the VAE's reconstruction loss in domain $A$ but also from the cycle and the GAN_AB losses in $\mathcal{L}^{II}$. As our experiments show, it is important to adapt these shared parts to $x$. This happens indirectly: during training the unshared layers of $E_B^U$ and $G_B^U$ are updated via the one-shot cycle loss (Eq. 11). Due to this change, all three loss terms in $\mathcal{L}^I$ are expected to increase and $G^S$ and $E^S$ are adapted to rectify this.

Selective backpropagation plays a crucial role in OST. Its aim is to adapt the unshared layers of domain $A$ to the shared representation obtained based on the samples of domain $B$. Intuitively, $L^I$ losses, which are formulated with samples of $B$ only, can be backpropagated normally, since due to the number of samples in $B$, $E^S$ and $G^S$ generalize well to other samples in this domain. Based on the shared latent space assumption, $E^S$ and $G^S$ would also fit samples in $A$. However, updating the layers of $G^S$ and $E^S$ based on loss $L^{II}$ (with selective backpropagation turned off, as is done in the ablation experiments of Tab. 1), would quickly lead to overfitting on $x$, since for every shared representation, the unshared layers in domain $A$ can still reconstruct this one sample. This increase in fitting capacity leads to an arbitrary mapping of $x$, and one can see that in this case, the mapping of $x$ is highly unstable during training and almost arbitrary (Fig. 2). If the shared representation is completely fixed at phase II, as in row 8 of Tab. 1, the lack of adaptation hurts performance. This is analogous to what was discovered in [25] in the context of adaptation in transfer learning.

Note that we did not add the cycle loss in the reverse direction. Consider the MNIST (domain $A$) to SVHN (domain $B$) translation (Fig. 3). If we had the cycle-loss in the reverse direction, all SVHN images (of all digits) would be translated to the single MNIST image (of a single digit) present in training. The cycle loss would then require that we reconstruct the original SVHN image from the single MNIST image (see rows 9 and 10 of Tab. 1).

### 3.3 Network Architecture and Implementation

We consider $x \in A$ and samples in B to be images in $\mathbb{R}^{3 \times 256 \times 256}$. We compare our results to state of the art method, CycleGAN [2] and UNIT [7] and use the architecture of CycleGAN, shown to be highly successful for a variety of datasets, for the encoders, decoders and discriminator. For a fair comparison, the same architecture is used when comparing OST to the CycleGAN and UNIT baselines. The network architecture released with UNIT did not outperform the combination of the UNIT losses and the CycleGAN architecture for the datasets that are used in our experiments.

Both the shared and unshared encoders (resp. decoders) consist of between 1 and 2 2-stride convolutions (resp. deconvolutions). The shared encoder consists of between 1 and 3 residual blocks after the convolutional layers. The shared decoder also consists of between 1 and 3 residual blocks before

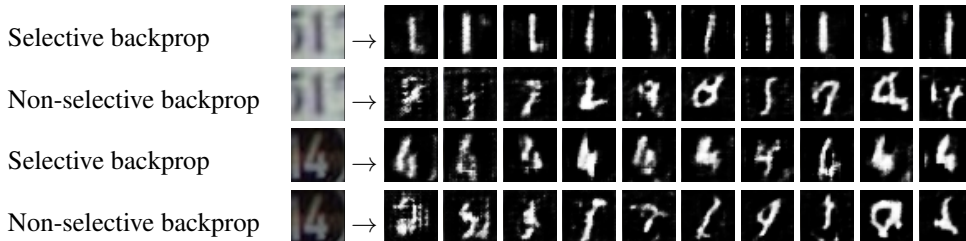

Figure 2: Mapping of an SVHN image to MNSIT. The results are shown at different iterations. Without selective backpropagation, the result is unstable and arbitrary.

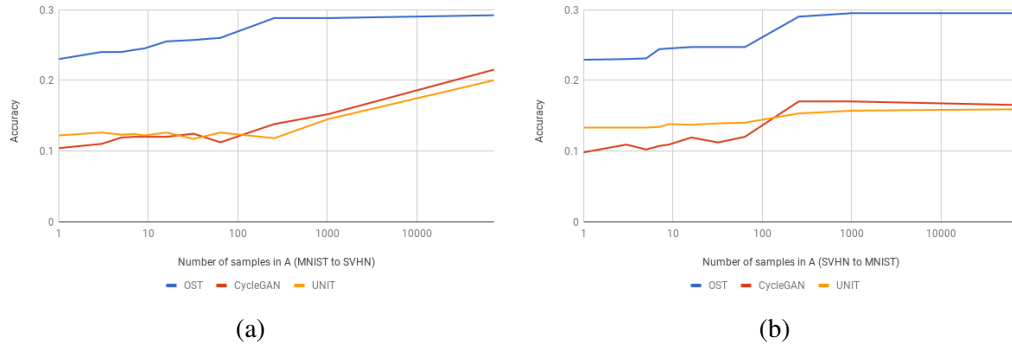

Figure 3: (a) Translating MNIST images to SVHN images. x-axis is the number of samples in $A$ (log-scale), y-axis is the accuracy of a pretrained classifier on the resulting translated images. The accuracy is averaged over 1000 independent runs for different samples. Blue: Our OST method. Yellow: UNIT [7]. Red: CycleGAN [2] . (b) The same graph in the reverse direction.

its deconvolutional layers. The number of layers is selected to obtain the optimal CycleGAN results and is used for all architectures. Batch normalization and ReLU activations are used between layers.

CycleGAN employs a number of additional techniques to stabilize training, which OST borrows. The first is the use of a PatchGAN discriminator [24], and the second is the use of least-square loss for the discriminator [23] instead of negative log-likelihood loss. For the MNIST [26] to SVHN [27] translation and the reverse translation, the PatchGAN discriminator is not used, and, for these experiments, where the input is in $\mathbb{R}^{3 \times 32 \times 32}$, the standard DCGAN [28] architecture is used.

## 4  Experiments

We compare OST, trained on a single sample $x \in A$, to both UNIT and CycleGAN trained either on $x$ alone or with the entire training set of images from $A$. We conduct a number of quantitative evaluations, including style and content loss comparison as well as a classification accuracy test for target images. For the MNIST to SVHN translation and the reverse, we conduct an ablation study, showing the importance of every component of our approach. For this task, we further evaluate our approach, when more samples are presented, showing that OST is able to perform well on larger training sets. In all cases $x$ is sampled from the training set of the other methods. The experiments are repeated multiple times and the mean results are reported.

**MNIST to SVHN Translation**   Using OST, we translated a randomly selected MNIST [26] image to an Street View House Number (SVHN) [27] image. We used a pretrained-classifier for SVHN, to predict a label for the translated image and compared it to the input MNIST image label.

Fig. 3(a) shows the accuracy of the translation for increasing number of samples in $A$. The accuracy is the percentage of translations for which the label of the input image matches that given by a pretrained classifier applied on the translated image. The same random selection of images was used for baseline comparison, and that accuracy is measured on the train images translated from A to B, and not on a separate test set. The reverse translation experiment was also conducted and shown in Fig. 3(b). While increasing the number of samples, increases the accuracy, OST outperforms the baselines even when trained on the entire training set. We note that the accuracy of the unsupervised mapping is lower than for the supervised one or when using a pretrained perceptual loss [21].

In a second experiment, an ablation study is conducted. We consider our method where any of the following are left out: first, augmentation on both the input image $x \in A$ and on images from $B$. Second, one way cycle loss, $\mathcal{L}_{cycle}$. Third, selective back propagation is lifted, and gradients from losses of $\mathcal{L}^{II}$ are passed through shared encoders and decoders, $E_s$ and $G_s$. The results are reported in Tab. 1. We find that selective back propagation has the largest effect on translation accuracy. One-way cycle loss and augmentation contribute less to the one-shot performance.

Table 1: Ablation study for the MNIST to SVHN translation (and vice versa). We consider the contribution of various parts of our method on the accuracy. Translation is done for one sample.

| Augment-ation | One-way cycle | Selective backprop | Accuracy (MNIST to SVHN) | Accuracy (SVHN to MNIST) |
|---|---|---|---|---|
| False | False | False | 0.07 | 0.10 |
| True | False | False | 0.11 | 0.11 |
| False | True | False | 0.13 | 0.13 |
| True | True | False | 0.14 | 0.14 |
| False | False | True | 0.19 | 0.20 |
| True | False | True | 0.20 | 0.20 |
| False | True | True | 0.22 | 0.23 |
| True | True | No Phase II update of $E^S$ and $G^S$ | 0.16 | 0.15 |
| True | Two-way cycle | True | 0.20 | 0.13 |
| True | Two-way cycle | False | 0.11 | 0.12 |
| True | True | True | **0.23** | **0.23** |

Table 2: (i) Measuring the perceptual distance [29], between inputs and their corresponding output images of different style transfer tasks. Low perceptual loss indicates that much of the high-level content is preserved in the translation. (ii) Measuring the style difference between translated images and images from the target domain. We compute the average Gram matrix of translated images and images from the target domain and find the average distance between them, as described in [29].

| Component | Dataset | OST | UNIT [7] | CycleGAN [2] | UNIT [7] | CycleGAN [2] |
|---|---|---|---|---|---|---|
| | Samples in $A$ | 1 | 1 | 1 | All | All |
| (i) Content | Summer2Winter | 0.64 | 3.20 | 3.53 | 1.41 | 0.41 |
| | Winter2Summer | 0.73 | 3.10 | 3.48 | 1.38 | 0.40 |
| | Monet2Photo | 3.75 | 6.82 | 5.80 | 1.46 | 1.41 |
| | Photo2Monet | 1.47 | 2.92 | 2.98 | 2.01 | 1.46 |
| (ii) Style | Summer2Winter | 1.64 | 6.51 | 1.62 | 1.69 | 1.69 |
| | Winter2Summer | 1.58 | 6.80 | 1.31 | 1.69 | 1.66 |
| | Monet2Photo | 1.20 | 6.83 | 0.90 | 1.21 | 1.18 |
| | Photo2Monet | 1.95 | 7.53 | 1.91 | 2.12 | 1.88 |

In another experiment, we completely froze the shared encoder and decoder in phase II. In this case, the mapping fails to produce images in the target distribution. In the SVHN to MNIST translation, for instance, the background color of the translated images is gray and not black.

**Style Transfer Tasks** We consider the tasks of two-way translation from Images to Monet-style painting [2], Summer to Winter translation [2] and the reverse translations. To asses the quality of these translations, we measure the perceptual distance [29] between input and translated images. This supervised distance is minimized in style transfer tasks to preserve the translation's content, and so a low value indicates that much of the content is preserved. Further, we compute the style difference between translated images and target domain images, as introduced in [29]. Tab. 2 shows that OST captures the target style in a similar manner to UNIT and CycleGAN when trained many samples, as well as CycleGAN trained with a single sample. While the latter captures the style of the target domain, it is unable to preserve the content, as indicated by the high perceptual distance. Sample results obtained with OST are shown in Fig. 4 and in the supplementary.

**Drawing Tasks** We consider the translation of Google Maps to Aerial View photos [24], Facades to Images [30], Cityscapes to Labels [31] and the reverse translations. Sample results are show in Fig. 4 and in the supplementary. OST trained on a single sample, as well as CycleGAN and UNIT trained on the entire training set obtain aligned mappings, while CycleGAN and UNIT trained on a single sample, either failed to produce samples from the target distribution or failed to create an

Table 3: (i) Perceptual distance [29] between the inputs and corresponding output images, for various drawing tasks. (ii) Style difference between translated images and images from the target domain. (iii) Correctness of translation as evaluated by a user study.

|   | Method | Images to Facades | Facades to Images | Images To Maps | Maps to Images | Labels to Cityscapes | Cityscapes to Labels |
|---|---|---|---|---|---|---|---|
| (i) | OST 1 | 4.76 | 5.05 | 2.49 | 2.36 | 3.34 | 2.39 |
|   | UNIT [7] All | 3.85 | 4.80 | 2.42 | 2.30 | 2.61 | 2.18 |
|   | CycleGAN [2] All | 3.79 | 4.49 | 2.49 | 2.11 | 2.73 | 2.28 |
| (ii) | OST 1 | 3.57 | 7.88 | 2.24 | 1.50 | 0.67 | 1.13 |
|   | UNIT [7] All | 3.92 | 7.42 | 2.56 | 1.59 | 0.69 | 1.21 |
|   | CycleGAN [2] All | 3.81 | 7.03 | 2.33 | 1.30 | 0.77 | 1.22 |
| (iii) | OST 1 | 91% | 90% | 83% | 67% | 66% | 56% |
|   | UNIT [7] ALL | 86% | 83% | 81% | 75% | 63% | 37% |
|   | CycleGAN [2] ALL | 93% | 84% | 97% | 81% | 72% | 45% |

aligned mapping. Tab. 3 shows that OST achieves a similar perceptual distance and style difference to CycleGAN and UNIT trained on the entire training set. This indicates that OST achieves a similar content similarity to the input image, and style difference to the target domain, as these methods. To further validate this, we asked 20 persons to rate whether the source image matches the target image (presenting the methods and samples in a random order) and list in Tab. 3 the ratio of "yes" answers.

## 5 Discussion

Being a one-shot technique, the method we present is suitable for agents that survey the environment and encounter images from unseen domains. In phase II, the autoencoder of domain $B$ changes in order to adapt to domain $A$. This is desirable in the context of "life long" unsupervised learning, where new domains are to be encountered sequentially. However, phase II is geared toward the success of translating $x$, and in the context of multi-one-shot domain adaptations, a more conservative approach would be required.

In this work, we translate one sample from a previously unseen domain $A$ to domain $B$. An interesting question is the ability of mapping from a domain in which many samples have been seen to a new domain, from which a single training sample is given. An analog two phase approach can be attempted, in which an autoencoder is trained on the source domain, replicated, and tuned selectively on the target domain. The added difficulty in this other direction is that adversarial training cannot be employed directly on the target domain, since only one sample of it is seen. It is possible that one can still model this domain based on the variability that exists in the familiar source domain.

## Acknowledgements

This project has received funding from the European Research Council (ERC) under the European Union's Horizon 2020 research and innovation programme (grant ERC CoG 725974). The contribution of Sagie Benaim is part of Ph.D. thesis research conducted at Tel Aviv University.

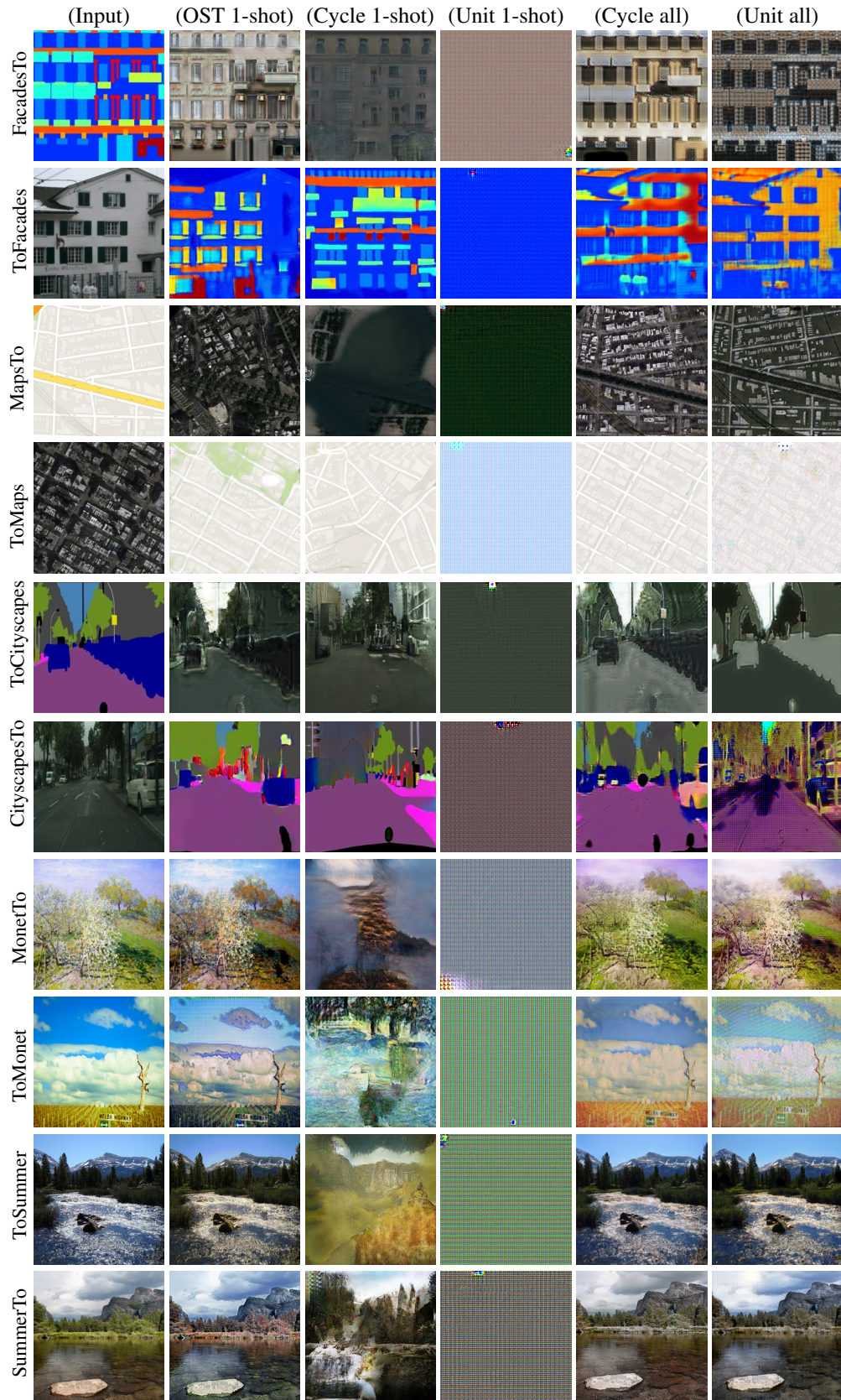

Figure 4: Translation for various tasks using OST (1 Sample), CycleGAN and UNIT (1 and Many Samples)

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
