[Supplementary Material]

# Supplementary Material for:
# One-Shot Unsupervised Cross Domain Translation

## A    More samples

The following figures show examples of the translation achieved by our method and the various baselines for images from the datasets we used in the experiments.

Figure 1: Additional mappings as given in Fig.3 of the paper for the task of Facades to Images and Images to Facades.

Figure 2: Additional mappings as given in Fig.3 of the paper for the task of Maps to Aerial View Images and Aerial View Images to Maps.

Figure 3: Additional mappings as given in Fig.3 of the paper for Cityscapes to Images and Images to Cityscapes.

Figure 4: Additional mappings as given in Fig.3 of the paper for Monet to Photo and Photo to Monet.

Figure 5: Additional mappings as given in Fig.3 of the paper for Summer to Winter and Winter to Summer.