[Reviews · NeurIPS 2018]

Reviewer 1



Authors propose an unsupervised domain translation approach for the one-shot case. While there are many proposed approaches for unsupervised domain translation authors claim that their approach is first to deal with the one-shot case. Across several collections and comparisons studies authors showcase that their approach (called OST) outperforms two existing approaches (UNIT and CycleGAN) in terms of translation accuracy, perceptual loss (mainly content and in some instances style difference) Overall I think that this work presents an interesting and very promising approach for performing unsupervised domain transfer in zero-shot instances when dealing with images. Author corroborate their claims through several studies which showcase the advantage of OST over two existing approaches. Questions/clarifications/suggestions: What about other cross domain translation tasks besides images? It would be good to compare the performance of OST on other data types. Is OST an abbreviation? Authors should tidy up their list of references. “NLP translation methods” - What other translation method are out there? For better clarity and readability Figure 1 should be shortened and instead authors should give more descriptive details of their approach in the introduction. For better clarity I would label the axis and introduce a legend in Figure 2. Users should give more details about the translation correctness study presented in table 3. Some typos: “The task we term” -> “the task that we term” “circulatory-loss” -> “circularity loss” “Can be serve” -> “can serve” “An horizontal” -> “a horizontal” “Cycle” -> “cycle” In several instances authors use commas rather than parentheses, e.g. “,after augmentation, to B using the transformation T,” -> (after augmentation, to B using transformation T)

Reviewer 2



The manuscript proposes a method for one-shot unsupervised cross domain translation by using a two-step process. The first step is to train a VAE for domain B, and the second step is to create a VAE for domain A using a sample in domain A. The experiments show that the proposed OST method perform similar with existing method with a lot of training samples in domain A. The problem itself is very interesting. Although the main idea is different from the existing methods UNIT and cycleGAN, the framework with VAE and GAN somehow follows UNIT and cycleGAN. The manuscript is overall well written, and the experimental results seems promising. My major concerns are as follows. 1. The framework shown in figure 1 is not clearly demonstrated, and the model is not clear enough, either. The manuscript seems only give a very rough idea and many details are unknown. For example, how the shared and unshared layers are defined? In the first step,E_B, G_B,and D_B can be trained. How the trained E_B, G_B and D_B are used in the second stage? How the G^S,E^S, E_B^U, E_A^U,G_B^U,G_A^U are trained in the second stage? 2. In the experiments, how is the accuracy in figure 2 and table 1 defined? The accuracy values seem very low. Is there any explanation? 3. Intuitively, why does the proposed OST method outperform UNIT or cycleGAN? Where does the advantage come from?

Reviewer 3



In overall, I think this paper proposes a well-designed two steps learning pipeline for one-shot unsupervised image translation. But the explanations about selective backpropagation in the rebuttal are still not clear to me. According to Eq.8-14, it seems that G^S and E^S are not updated in phase II. But according to the Tab. 1 and the rebuttal, they seem to be selectively updated. I strongly suggest the authors to explain the details and motivation in the method part if this paper is accepted. I keep my initial rating. =========================================== This paper proposes a two steps method for one-shot unsupervised image-to-image translation task. This method can help to enable the one-shot unsupervised image translation and also improve the performance continuously given more training samples. Pros: * The proposed idea is interesting that designing a two steps learning pipeline based on the shared latent space assumption which is implemented with the simple weight sharing technique. * The writing is easy to follow, and empirical experiments show that it is able to translate the image in one-shot setting. Cons: * If I understand correctly, the shared layers of G^S and E^S will only be updated with loss L^I in phase two of training. The explanation for such selective backpropagation in line 146-148 is not so convincing to me. ** I would like the authors to explain the function and motivation for such selective backpropagation in more details, since it is the key for the one-shot unsupervised image-to-image translation. ** Does the training of G^S and E^S aim at keeping the shared latent constraint? ** Why only use one-way cycle loss? I would like the authors to explain the reason of not using two-way cycle loss and show the comparison. * In the results of style transfer and drawing tasks, the authors only show the OST 1-shot results. I wonder how the proposed method will perform on these tasks with all-shot setting, since for MNIST to SVHN Translation OST is able to perform better than CycleGAN and UNIT when more samples are presented.